# Positional Behavior and Substrate Use in Wild Tibetan Macaques

**DOI:** 10.3390/ani12060767

**Published:** 2022-03-18

**Authors:** Peng-Hui Li, Wen-Bo Li, Bo-Wen Li, Ya-Dong Li, Xi Wang, Jin-Hua Li

**Affiliations:** 1School of Resources and Environmental Engineering, Anhui University, Hefei 230601, China; ahupenghuili@126.com (P.-H.L.); ahuwenboli@126.com (W.-B.L.); libowen0116@126.com (B.-W.L.); ydli266@163.com (Y.-D.L.); xwang@ahu.edu.cn (X.W.); 2International Collaborative Research Center for Huangshan Biodiversity and Tibetan Macaque Behavior Ecology, Hefei 230601, China; 3School of Life Sciences, Hefei Normal University, Hefei 230601, China

**Keywords:** Tibetan macaque, positional behavior, substrate use, individual development, terrestriality

## Abstract

**Simple Summary:**

Body size and individual development are essential factors that affect primate movement through the canopy and access to food. However, only a few studies have been conducted on immature wild macaques. Although adults are significantly heavier than the other age groups, this study found that they did not exhibit higher frequencies of climbing and bridging. Jumps and suspensions were more frequent in juveniles than adults. We also found that juveniles exhibited rare behaviors during play, such as cling locomotion, suspensory locomotion, and bounding. We hypothesized that juveniles would exhibit diverse positional behaviors associated with altered skeletal muscle development. Diverse positional abilities facilitate habitat exploitation and the avoidance of danger.

**Abstract:**

Body size and individual development significantly affect positional behavior and substrate use. However, only a few studies have been conducted on immature wild macaques. We studied wild Tibetan macaques (*Macaca thibetana*) inhabiting Mt. Huangshan, China, to explore the degree of interspecific variation in positional behavior in relation to body weight and individual development. From September 2020 to August 2021, we used instantaneous scan sampling (duration 5 min, interval 10 min) to record age–sex groups, locomotions, postures, and substrate attributes. The results showed that Tibetan macaques used terrestrial substrates in nearly two-thirds of the recorded observations. The main postural modes were sitting and quadrupedal standing. The main locomotor modes were quadrupedal walking and climbing among all age–sex group records. Positional behavior and substrate use in adults only significantly differed from those in juveniles and infants. Although adult males were larger than the other age–sex groups, they did not climb and bridge more frequently than the other age–sex groups. The frequency of climbing, leaping, and suspension was significantly higher in juveniles than in adults. In addition, adult males used terrestrial and larger substrates more frequently, while juveniles and infants used arboreal substrates and terminals more frequently than adult males during traveling and feeding. We hypothesize that the more positional behavioral spectrum of Tibetan macaque juveniles’ may be related to rapid skeletal muscle development. These results suggest that differences in interspecific positional behavior may be caused by the individual development and survival needs of individuals, rather than just body size.

## 1. Introduction

Positional behaviors are behavioral patterns that have evolved in wild animals to solve problems, such as obtaining food resources, crossing obstacles, and avoiding predators in their habitat [1,2]. Primates have evolved morphological features that increase grasping and mobility and utilize terminal branches using their graspable hands, feet, and tails [3,4,5]. These morphological features have led to the development of positional behavioral abilities to adapt to terrestrial and arboreal substrates, such as tail suspension, forelimb suspensory locomotion, vertical climbing, and fist quadruped walking [6]. Therefore, the relationship between primate positional behavior, morphology, and ecology is essential for understanding their adaptive evolution [7,8,9].

Currently, three views have been formed on the influence of morphological characteristics, such as body size, relative tail length, and intermembral index (IMI), on positional behavior and substrate use [10,11,12,13]. First, larger primates are more inclined to adopt more conservative ways of crossing gaps, such as climbing and suspension, whereas smaller primates adopt more leaping behaviors [14,15,16]. For example, Fleagle et al. studied seven New World monkeys in Suriname and found that the frequency of climbing increased with body mass, and the frequency of leaping increased with reduced body mass [10]. Larger primates used larger substrates more frequently than smaller primates, and smaller primates used smaller substrates. For example, heavier Assamese macaques (*Macaca assamensis*) utilize branches more frequently than rhesus monkeys (*Macaca mulatta*) during movement, and rhesus monkeys utilize twigs more frequently than Assamese macaques [13]. Second, the IMI, representing the ratio of forelimb to hindlimb length, is a good predictor of locomotor tendency. Primates with lower IMI have better leaping ability, primates with higher IMI have better suspension and climbing abilities, and primates with intermediate IMI have better quadrupedal locomotion ability [17]. Third, the tail plays an important role in maintaining body balance, and primates with relatively longer tails have better leaping ability [11,12,17]. However, the prediction of positional behavior by external morphological characteristics is not always consistent [18,19], and additional studies are needed to refine it. The influence of skeletal musculature, locomotor skills, and ecological factors on positional behavior at different life-history stages needs to be studied in greater detail [20,21,22].

Individual development also influences the expression of positional behavior [1,22,23,24]. Immature macaques cannot be summarized simply as “small individuals with weak skeletal muscle strength and poor neural control of limb movements” [19]. The anatomy of primates has shown that physical features, such as musculoskeletal development and the center of mass, constrain positional behavior in different ways during individual development [25,26,27]. Some primates show positive anisotropic growth of the forearm extensors and forearm flexors early in individual development [28]. Fast skeletal muscle development provides a mechanical advantage to immature individuals, enabling them to generate greater propulsive forces than adults during limb acceleration and body movement [28]. The biomechanical consequences of changing weight distribution during development in olive baboons (*Papio anubis*) are consistent with the developmental expression of positional behavior [27]. These developmental patterns may cause remarkable differences in positional behavior between age classes [21]. For example, red-legged white-armed langurs (*Pygathrix nemaeus*) exhibit 50 positional patterns as juveniles, compared to only 23 as adults [19]. Sarringhaus et al. (2014) observed that the number of sublocomotor patterns in chimpanzees (*Pan troglodytes*) from infancy to adulthood decreased (infants—29, juveniles—25, young adults—20, and adults—11) [29]. Immature primates appear to exhibit a more diverse positional behavior spectrum. However, some researchers consider positional behavior to be conservative, even though it develops earlier [30,31]. Despite differences in activity budgets and support use in juvenile, subadult, and adult black and white colobus monkeys (*Colobus angolensis palliatus*), their locomotor profiles are remarkably consistent [21]. Therefore, the extent to which intraspecies variation in positional behavior arises from differences in body weight, substrate use, and musculoskeletal development needs to be analyzed.

This study focused on the positional behavior and substrate use of wild Tibetan macaques (*Macaca thibetana*), a *Cercopithecidae* primate endemic to China [32]. They often live in mixed evergreen broadleaf–deciduous broadleaf forests and inhabit low woodland and scrubland [33]. Tibetan macaques are found at altitudes of 300 m to 2100 m [34,35,36]. Therefore, multiple positional abilities are required for adaptation to the ecological environment. Tibetan macaques are the largest monkeys in the *Macaca* genus [32]. Significant differences in body weight were found between different age–sex groups, with infants weighing 2.5 kg, juveniles weighing 4.75 kg, subadult females weighing 8 kg, subadult males weighing 12.7 kg, adult males weighing 18.3 kg, and adult females weighing 12.8 kg [35]. The growth rate of the limb bones in Tibetan macaque juveniles is greater than their weight gain [37]. Only a few anatomical studies have suggested that Tibetan macaques are terrestrial quadrupedal primates. However, quantitative studies on the positional behavior and substrate use of Tibetan macaques in the natural environment are lacking. It is unclear whether Tibetan macaques are terrestrial, and it is also unclear how positional behavior and substrate use differ between age–sex groups. 

Given that this is a preliminary study of the positional behavior and substrate use of Tibetan macaques, we proposed the following predictions. Prediction 1: As adult males have a greater body mass, we suggest that adult males climb more and leap less than adult females, subadults, juveniles, and infants. Prediction 2: If body size is a good predictor of patterns of substrate use, we suggest that adult males use more ground and substrates with larger diameters than adult females, subadults, juveniles, and infants. Prediction 3: Owing to rapid skeletal muscle development in infants and juveniles, we expect juveniles to exhibit more positional behavior.

## 2. Methods

### 2.1. Study Site 

Our study was conducted at the Niejiashan Research Base (NRB) at Mt. Huangshan, Anhui Province, eastern-central China (30°12′ N, 18°27′ E, 250–650 m above sea level), founded by the International Collaborative Research Center for Huangshan Biodiversity and Tibetan Macaque Behavioral Ecology, Anhui University, in 2017. Monkeys were fitted with global positioning system (GPS) collars to track their movement, and we found that their movement spanned three regions: Tanjiaqiao Town, Tangkou Town, and Huangshan Mountain, with elevations ranging from 260 m to 1100 m (Figure 1). The vegetation in this area is subtropical and vertically distributed. *Pinus massoniana* forest is present at an altitude of 200–400 m; evergreen broad-leaved forest, mixed evergreen–broad-leaved, and deciduous broad-leaved forests are present at 400–600 m; and mixed evergreen–broad-leaved and deciduous broad-leaved forests are present at 600–1200 m [34]. Common tree species include *Pinus massoniana*, *Cunninghamia lanceolata*, *Phyllostachys heterocycle*, *Castanopsis eyrei*, *Castanopsis sclerophylla*, and *Liquidambar formosana*, and common shrub species include *Theaceae* and *Rhododendron simsii*. There are also many cliffs, exposed rocks, and streams.

### 2.2. Study Groups

We selected a group of wild Tibetan macaques (THII) close to the study site for observation (*n* = 30). We classified the Tibetan macaques into five age–sex groups: adult males, adult females, subadults, juveniles, and infants. Although infants are not entirely independent of their mothers, they gradually develop independent locomotions and postures with age; therefore, we analyzed their behavior. Subadults are similar to adults in body size, but their physiological state is not mature and there are differences in social behaviors, thus distinguishing subadult individuals from juveniles. In addition, there is apparent sexual dimorphism in Tibetan macaques; therefore, we analyzed adult males and adult females separately. The age–sex classes were determined based on differences in body size, hair color, facial color, reproductive organs, and behavior [32]. The group consisted of six adult males, eight adult females, three subadults, seven juveniles, and six infants. In December, we suspected that one adult male died of a disability; therefore, his behavior was not analyzed.

### 2.3. Behavioral Data Collection

Although the research team had been tracking wild monkeys for two years, we fitted separate GPS collars (HQAN40S, GLOBAL MASSAGER) to one adult male and one adult female in the group to improve the efficiency of our observations. Before data collection, we conducted pre-experiments on the positional behavior and substrate use of Tibetan macaques from July to September 2020. The results showed that the monkeys often used five locomotors and six postural modes. The observers were trained to become proficient in visually detecting the height, diameter, and inclination of substrates. Data on positional behavior and substrate use were collected by one investigator. We used instantaneous scan sampling to record the data from September 2020 to August 2021. This sampling method is the standard for studying positional behavior and ensuring data independence [38,39]. Despite the habituation of the monkeys, the dense scrub was not conducive to observation; therefore, instantaneous scan sampling was performed for a 5-min duration with 10-min intervals. Each scan sampling was performed from left to right to record all visible individuals within 5 min. We quickly determined the age–sex class, activity, main locomotion or posture, and substrate use when scanning individuals. When an individual was scanned for feeding or resting, their posture was recorded (Table 1). When individuals were observed traveling, their locomotor behavior was recorded (Table 1). We also visually measured the height and diameter of the substrates used for all behaviors (Table 1). All behaviors are defined in Table 1 [7,39,40,41].

Before starting each day’s tracking, we checked the monkeys’ nocturnal sites from the previous night using GPS collars. Observations throughout the day began at 06:00 a.m. and ended when the monkeys entered their sleep sites. We also collected half-day data beginning from the first encounter and ending when the monkeys disappeared for more than an hour or entered their sleep sites. We used binoculars to observe monkeys within a 5 m distance. Subsequently, the effective scanning period for the annual positional behavior study was 72 days, with an average of 6 days per month (2–12 days/month, SD = 2.75; *N* = 12 months). The total number of scans for the year was 1544, and the average number per month was 129 (21–218 scans/month, SD = 55.90; *N* = 1544). The number of individual behaviors for the year was 13,751, with an average number of 1146 per month (122–2327 numbers/month, SD = 601.12; *N* = 13,751).

### 2.4. Data Analysis

We treated each individual behavior scanned as an independent sample. To summarize the statistics, we expressed the monthly utilization of each behavior using the monthly contribution of each locomotor mode, postural mode, and substrate category divided by their corresponding mode’s total contribution. We used the mean of the monthly utilization of each behavior as the pattern of positional behavior and substrate use for Tibetan macaques throughout the year [13,39]. We also calculated the percentage of different monthly behaviors for the different age and sex groups. We tested all age–sex group data for normality using the Kolmogorov–Smirnov test and homogeneity of variances using Levene’s test. If the assumption was violated, we performed a non-parametric Kruskal–Wallis test to test for differences in utilization of each behavior across age–sex groups [39,42]. We also compared the differences between each two groups using multiple comparisons in the nonparametric Kruskal-Wallis test and adjusted for significant values using the Bonferroni correction. We dropped the locomotor modes of cling locomotion, dropping, and suspensory locomotion, as there were insufficient records of these behaviors. We combined vertical climbing and clambering into the category “climbing”. We performed all statistical analyses using SPSS software version 26.0.0.0, with the significance level set at *p* ≤ 0.05 and the confidence limit set at 95%. All figures were drawn using OriginPro 2021b SR2 version 9.8.5.212 software.

### 2.5. Ethics Statement

This study complied with the regulations of the Chinese Institutional Ethics Committee on animal ethics. This study was authorized and approved by the China Wildlife Administration. The Huangshan Garden Forestry Bureau in China permitted us to conduct a local field study.

## 3. Results

### 3.1. Description of Positional Behavior and Substrate Use 

The mean values for the monthly percentages of positional behavior and substrate use records are shown in Table 2. The main locomotor modes were quadrupedal walking (57.9%) and climbing (24.9%), followed by quadrupedal running (7.8%), leaping (7.1%), and bridging (2.7%). The main postural modes were sitting (85.8%), quadrupedal standing (9%), and included low-frequency lying, clinging, suspension, and bipedal standing. They used terrestrial substrates in nearly two-thirds of the recorded observations, while branches were the main arboreal substrates used (Table 2).

### 3.2. Age- and Sex-Based Differences in Positional Behavior

Infants, juveniles, subadults, adult females, and adult males exhibited similar locomotor repertoires, but different frequencies of particular locomotor behaviors (Kruskal–Wallis test, quadrupedal walking: χ2 = 27.293, df = 4, *p* = 0.000; climbing: χ2 = 28.090, df = 4, *p* = 0.000; leaping: χ2 = 14.635, df = 4, *p* = 0.006; bridging: χ2 = 10.722, df = 4, *p* = 0.03). Overall, the predominant locomotor behaviors for all age–sex groups were quadrupedal walking and climbing while traveling, regardless of whether this occurred on the ground or in the canopy. During travel, infants and juveniles climbed significantly more than adults, and adult males had the highest frequencies of quadrupedal walking. Juveniles leaped more frequently than adults. There were no significant differences in quadrupedal running (Kruskal–Wallis test, χ2 = 2.480, df = 4, *p* = 0.648) (Figure 2).

During feeding, the different age–sex groups mainly sat and stood quadrupedally. Infants and adult females did not exhibit a cling foraging posture. Juveniles engaged in lying more frequently than adult females (*p* < 0.05). The suspension foraging posture decreased with age until it disappeared entirely in adulthood (Figure 3).

### 3.3. Age- and Sex-Based Differences in Substrate Use

All age–sex groups traveled and fed in all substrate height and size categories. During travel, different age–sex groups exhibited significant differences in the frequency on the ground, lower forest strata, and middle forest strata (Kruskal–Wallis test, ground: χ2 = 29.604, df = 4, *p* = 0.000; lower forest strata: χ2 = 31.867, df = 4, *p* = 0.000; middle forest strata: χ2 = 16.687, df = 4, *p* = 0.02), but not in the upper forest strata (χ2 = 9.453, df = 4, *p* = 0.51). Adults used the terrestrial substrates significantly more often than juveniles and infants (adult males and females: 80.83% and 76.03% vs. juveniles and infants: 41.61% and 43.04%, respectively); juveniles and infants were significantly more likely to move in the lower forest strata than adults (adult males and adult females: 10.13% and 10.46% vs. juveniles and infants: 29.37% and 30.75%, respectively); all categories were less likely to move in the upper forest strata. Age–sex groups differed significantly in the use of twigs and boughs during traveling (Kruskal–Wallis test, twigs: χ2 = 10.240, df = 4, *p* = 0.037; boughs: χ2 = 10.363, df = 4, *p* = 0.035), while the frequency of branch use did not differ significantly (Kruskal–Wallis test, χ2 = 3.389, df = 4, *p* = 0.495). During travel, infants and juveniles used twigs more frequently than adults, and adult males used boughs more frequently than juveniles and infants. All categories preferred to move on the branches (Table 3).

The frequency of foraging on the ground was significantly different between age and sex groups (Kruskal–Wallis test, χ2 = 10.601, df = 4, *p* = 0.031) and there were no significant differences between the lower, middle, and upper forest strata (lower forest strata t: χ2 = 5.229, df = 4, *p* = 0.265; middle forest strata: χ2 = 5.704, df = 4, *p* = 0.222; upper strata χ2 = 8.119, df = 4, *p* = 0.087). Adult males spent significantly more time foraging on the ground than juveniles and infants (adult males 59.53% vs. juveniles and infant monkeys: 31.63% and 31.12%), and immature individuals foraged more often in trees, but no significant differences existed. Different age groups showed significant differences in the use of twigs and branches (Kruskal–Wallis test, twigs: χ2 = 30.000, df = 4, *p* = 0.000; branches: χ2 = 25.642, df = 4, *p* = 0.000), but not in the use of boughs (Kruskal–Wallis test, χ2 = 5.475, df = 4, *p* = 0.242). Infants and juveniles foraged significantly more frequently than adults and subadults foraged on twigs, and adults and subadults foraged significantly more frequently using branches than juveniles and infants. All categories spent less time foraging in the upper forest strata (above 10 m) and boughs (Table 4).

## 4. Discussion

### 4.1. Positional Behavior of Tibetan Macaques

This study discusses the positional behavior and substrate use of wild Tibetan macaques living at low elevations in Mt. Huangshan, China. In nearly two-thirds of the recorded observations, Tibetan macaques used terrestrial substrates, avoiding substrates above 10 m (Table 2). Some researchers consider a species terrestrial if it spends more than 60% of its time on the ground [39,43,44]. A previous study had classified Japanese macaques (*Macaca fuscata*) as terrestrial (68.3% in the terrestrial substrate) [12] based on the same criteria. Thus, Tibetan macaques are terrestrial primates. 

The primary locomotor behaviors of the Tibetan macaque were quadrupedal walking and climbing, and the primary postural behaviors were sitting and quadrupedal standing (Table 2). During arboreal travel, *M. thibetana* crossed gaps in the canopy by climbing, grasping small supports with their forelimbs, and lowering their center of gravity by bending down their hindlimbs, which is an essential way for large primate arboreal movements to maintain balance [45]. Climbing accounts for almost a quarter of the entire locomotor repertoire, significantly more than in Japanese macaques, rhesus macaques, and long-tailed macaques (*Macaca fascicularis*), and less than in Assamese macaques [12,13,46]. The positional profiles of Tibetan macaques are consistent with the skeletal anatomy. Tibetan macaques exhibit lower growth rates of the hindlimbs and relatively longer forelimbs [37]. Pan et al. (1989) considered this growth pattern to have been evolved to adapt to climb during traveling and foraging [47].

### 4.2. Effect of Body Mass on Intraspecific Positional Behavior and Substrate Use

As adult Tibetan macaques have a greater body mass, we expected that adult males would climb more and leap less than adult females, subadults, juveniles, and infants. Although leaping behavior was as predicted, juveniles and infants climbed more frequently than adults (Figure 2). Larger primates use more energy to climb than smaller primates. Juveniles and infants used twigs and arboreal support more often than adult males. Foraging and movement on the periphery of the canopy may force animals to increase their frequency of climbing and bridging [18,48]. Adult males are more inclined to walk quadrupedally on coarse and terrestrial substrates, most likely to conserve energy [31,49]. We also found no differences in positional behavior between adults and subadults, suggesting that Tibetan macaques adopt a similar approach to support body weights between 12 and 18 kg in body size. Some researchers have argued that, due to the morphological limitations of individual development, it is challenging to discover differences in positional behavior between primate subadults and adults [21,50]. Therefore, Prediction 1 was not supported.

Given the discontinuity of the arboreal support and the fragility of the terminal branches [10,21], we expected adults to be more active on the ground and coarse branches, and that other age–sex groups would be more active on trees and twigs. Infants and juveniles foraged significantly more frequently than adults, and subadults foraged on the terminal branches; however, there were no differences between adults and subadults. Therefore, Prediction 2 was not supported. Tibetan macaques are cautious about exploiting terminal twigs once their body weight exceeds 12 kg. Zhu et al. (2015) suggested that immature golden snub-nosed monkeys (*Rhinopithecus roxellana*) occupy higher strata for predator avoidance [31]. Historically, there were also many terrestrial carnivores on Mt. Huangshan, including *Canidae* (e.g., *Canis lupus* and *Cuon alpinus*), *Felidae* (e.g., *Neofelis nebulosa* and *Panthera pardus*), and *Ursidae* (e.g., *Ursus thibetanus*) [51]. However, predator avoidance is episodic and does not have a significant life history. Terminal branches have more young leaves and fruits and are more easily digested by juveniles [3,52]. Tibetan macaque subadults always prefer grooming and presenting rumps to high-ranking adults to enhance their social status, leading to a similar substrate pattern. These patterns suggest that substrate preference may be related to diet, social behavior, and body size.

### 4.3. Ontogeny of Positional Behavior in Tibetan Macaque

Many attempts have been made to explain the development of positional behavior in primates in the laboratory regarding skeletal muscle changes during individual development [53,54]. However, the positional abilities of immature individuals tested in the laboratory are challenging to express in the wild [53]. Wild Tibetan macaque infants showed a positional repertoire similar to that of adult females, none of which exhibited a cling foraging posture. Similarly, *Macaca*, *Alouatta*, and *Semnopithecus* monkeys acquired similar positional abilities as adults earlier. Turnquist et al. (1994) suggested that locomotor and postural abilities in primates change dramatically during the first postnatal year [24]. Tibetan macaque infants exhibited a higher frequency of suspension postures than other age groups. The suspension posture extends the body’s range, which is essential for small primates.

Our results support prediction 3 that Tibetan macaque juveniles exhibit a more diverse positional spectrum than adults. Juveniles exhibited a cling foraging posture that was absent in infants, and a hindlimb suspension foraging posture that was absent in adults. We also found that juveniles exhibited rarer behaviors during play, such as quadrupedal suspension, cling locomotion, suspensory locomotion, and bounding. The diversity of positional behaviors in juveniles may be related to musculoskeletal development. At 2–3 years postnatal, the increase in the bone length of the anterior and posterior limbs of Tibetan macaques is faster than the increase in body weight [37]. Smaller bodies, flexible limbs, and rapidly developing muscles may allow juveniles to gain greater propulsive and pulling forces than adults [28,31,54,55]. Therefore, we observed that Tibetan macaque juveniles exhibited a higher frequency of leaping, suspensory locomotions, and bounding. Some researchers also believe that play allows the development and practice of fine motor skills [26,56]. Wunderlich et al. referred to juveniles as “ecological adults” because they must address ecological challenges similar to those of adults, despite their small bodies and incomplete musculoskeletal development [57]. Diverse positional capabilities can be used to address ecological challenges, such as foraging, anti-predation, and avoidance of attack.

The frequency of the Tibetan macaques’ behavior below and to the side of the substrate decreased with age. There were no significant differences in positional behavior between subadult and adult Tibetan macaques. The morphological limitations of the skeletal muscles may be the main reason for this similarity.

## 5. Conclusions

Tibetan macaques are terrestrial primates. A higher frequency of climbing behavior can be adapted for foraging in low scrub brush. Changes in body mass and skeletal muscle due to individual development allow immature Tibetan macaques to develop more positional behaviors, such as leaping, suspension, and clinging locomotion. These behaviors facilitate the exploitation of habitats by immature Tibetan macaques more than adults. However, we did not measure the forces generated by the skeletal muscles of Tibetan macaques at different life stages. Future studies should specifically analyze the relationship between the morphometrics of the skeleton and the forces required for different positional behaviors, which may provide a basis for the positional behaviors of primate adults not differing from those of subadults.

## Figures and Tables

**Figure 1 animals-12-00767-f001:**
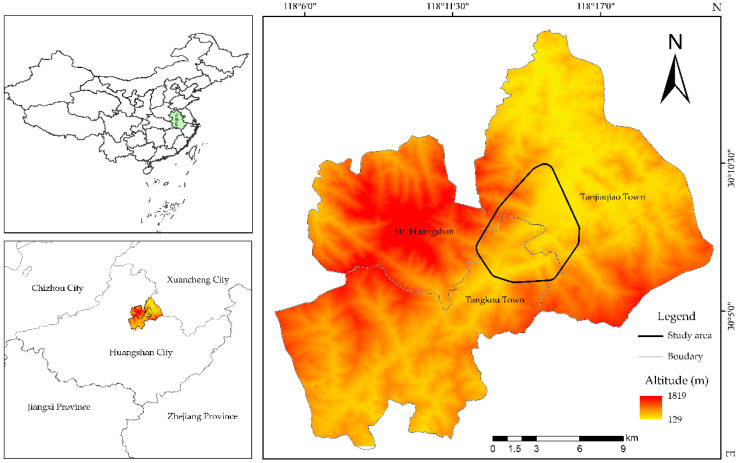
Range of the study area.

**Figure 2 animals-12-00767-f002:**
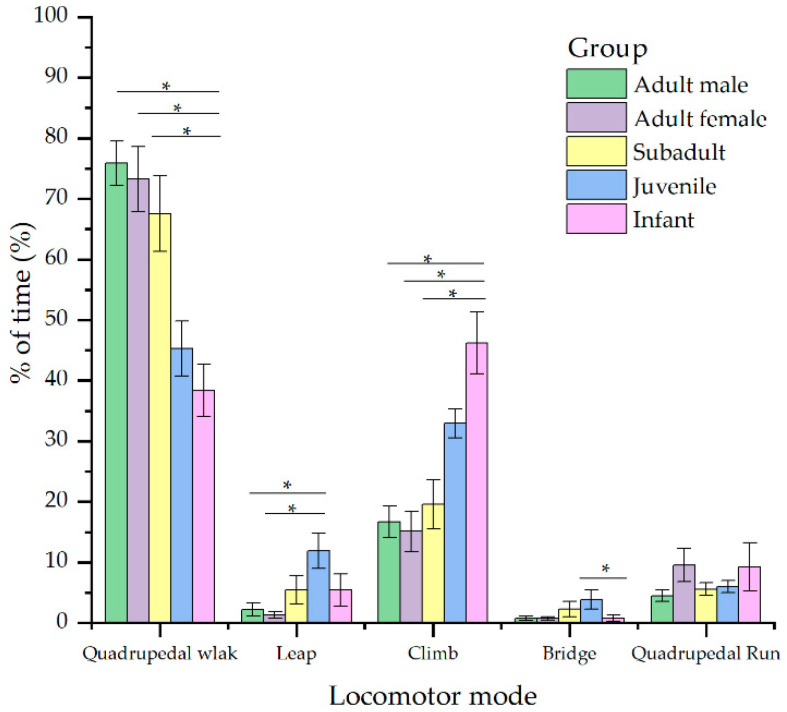
Age- and sex-based differences in locomotor behavior during traveling. * means significant difference (*p* < 0.05).

**Figure 3 animals-12-00767-f003:**
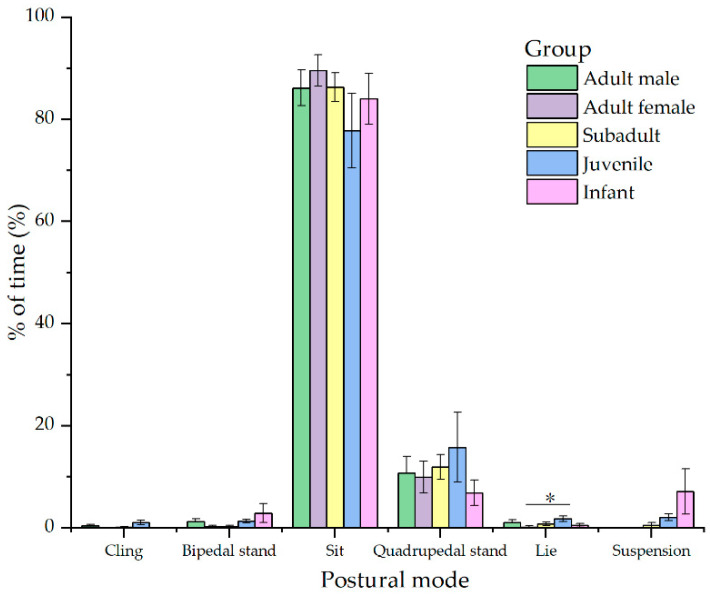
Age- and sex-based differences in postural behavior during feeding. * means significant difference (*p* < 0.05).

**Table 1 animals-12-00767-t001:** Variables and definitions of activity, positional behavior, and substrate use in Tibetan macaques.

Terms	Definitions *
**Activity**	
Feeding	Searching for, chewing, and swallowing food
Traveling	Changing the position of the body in space by walking, running, leaping, climbing, etc., and not obtaining food within 5 s
Resting	Maintaining a stationary position without movement
Social grooming	One individual uses their fingers or palms to separate and smooth the hairs of another individual and occasionally picks up certain small particles from the separated hairs or exposed skin and puts them in their mouth to chew
Other	Play and unusual behaviors, including sexual behavior or aggression
**Locomotor mode**	
Quadrupedal Walking	Movement in a particular gait along a substrate with an inclination of less than 50°
Climbing	Quadrupedal movement on a large inclination (substrate angle > 50°) or unstable substrate
Leaping	Crossing the substrate gap involving free flight movements in which the hind limbs provide propulsion
Quadrupedal running	Similar to quadrupedal walking, but faster and with a brief period of free flight in the air (i.e., all limbs off the ground)
Bridging	Movement to cross the substrate gap with the hind limbs grasping one side of the substrate and the forelimbs grasping the other side of the substrate as the body slowly moves towards the front
**Postural mode**	
Sitting	Relies primarily on the ischia to support most of the body’s weight
Quadrupedal standing	Relies primarily on the front and hind limbs to support most of the body weight, with the trunk horizontal
Bipedal standing	Relies primarily on the hind limbs to support most of the bodyweight, upright or bent, with the forelimbs sometimes touching the substrates.
Lying	The torso rests relatively horizontally above the substrate, primarily supporting the weight
Suspension	The forelimbs or hindlimbs grip the substrate firmly to support the body, and the rest of the body does not touch the substrate, and the torso is relatively extended
Clinging	Body against a vertical substrate with hands gripping or holding the substrate firmly to maintain stability
**Substrate height**	
Ground	Substrate with only 0 m vertical height
Lower strata	0–5 m height above ground
Middle strata	5–10 m height above ground
Upper strata	Height above 10 m above ground
**Substrate size**	
Twigs	Terminal branches usually smaller than 2 cm in diameter
Branches	Monkeys can usually grasp branches between 2 cm and 10 cm in diameter
Boughs	Substrate diameter greater than 10 cm

* Definitions of activities were adapted from Mekonnen et al. (2018) [39], locomotor and postural modes from Hunt et al. (1996) [40], substrate height from Fan et al. (2013) [41], and substrate di-ameter from Mittermeier (1978) [7].

**Table 2 animals-12-00767-t002:** Percentages of records of locomotor and postural modes and substrate use for Tibetan macaques.

	Mean	SD
**Locomotor mode**		
Climbing	24.9	6.3
Bridging	2.7	2.8
Quadrupedal running	7.8	4.0
Quadrupedal walking	57.9	9.4
Leaping	7.1	5.6
**Postural mode**		
Clinging	2.8	1.4
Bipedal standing	1.0	0.7
Quadrupedal standing	9.0	4.8
Suspension	0.8	0.7
Siting	85.8	4.9
Lying	1.5	0.9
**Substrate height**		
Ground	61.3	10.9
Lower forest strata	18.9	4.4
Middle forest strata	12.8	5.6
Upper forest strata	7.0	5.9
**Substrate size**		
Twigs	8.3	3.9
Branches	55.5	5.1
Boughs	36.2	5.1

Abbreviations: Mean, mean values for monthly percentages of records; SD, standard deviation; mean values for monthly percentages of records.

**Table 3 animals-12-00767-t003:** Substrate use (%) of age–sex groups during traveling.

	Adult Males	Adult Females	Subadults	Juveniles	Infants
Substrate height					
Ground	80.83	76.03	61.57	41.61 ^a,b^	43.04 ^a,b^
Lower	10.13	10.46	14.70	29.37 ^a,b^	30.75 ^a,b^
Middle	8.20	9.13	21.00	24.74 ^a,b^	23.19
Upper	0.83	4.38	2.73	4.82	3.00
Substrate size					
Bough	24.24	19.29	12.65	11.72 ^a^	10.32 ^a,b^
Branch	58.47	63.39	61.45	58.17	48.78
Twig	17.29	17.32	25.90	30.11 ^a,b^	40.90 ^a,b^

Note: ^a^ indicates a significant difference from an adult male and ^b^ indicates a significant difference from an adult female, which had been adjusted for significance by Bonferroni correction.

**Table 4 animals-12-00767-t004:** Substrate utilization (%) by age–sex groups during feeding.

	Adult Males	Adult Females	Subadults	Juveniles	Infants
Substrate height					
Ground	59.53	45.57	43.32	31.63 ^a^	30.12 ^a^
Lower	19.55	28.25	25.25	30.42	37.10
Middle	13.36	18.62	23.39	28.78	29.00
Upper	7.56	7.56	8.04	9.17	3.55
Substrate size					
Bough	2.94	0.51	3.26	1.26	2.65
Branch	59.72	61.73	67.73	40.16 ^a,b,c^	23.96 ^a,b,c^
Twig	37.34	37.77	29.01	58.57 ^a,b,c^	73.39 ^a,b,c^

Note: ^a^ indicates a significant difference from an adult male, ^b^ indicates a significant difference from an adult female, and ^c^ indicates a significant difference from a subadult, which had been adjusted for significance by Bonferroni correction.

## Data Availability

The data supporting the results of this study are available upon request from the corresponding authors. The data are not publicly available due to privacy theory restrictions.

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
