# Peer review of "Positional Behavior and Substrate Use in Wild Tibetan Macaques"

_animals, 2022, doi:10.3390/ani12060767_

Round 1

Reviewer 1 Report

In general, the text is very well written. The authors provide novel information that is relevant for comparative purposes within the genus Macaca. The information follows a coherent order. Nevertheless, the first component of the manuscript lacks further theoretical information about the relationship between ontogeny and positional behavior. This is a core issue in this manuscript. This issue is also relevant to be indicated in order to provide a background of the predictions to be tested. In this sense, I congratulate the authors for presenting predictions to be tested in the article rather than just listing objectives to be studied as usually presented in this kind of study. The methods are well-explained and the statistical tests are adequate.

In the discussion, the results regarding Prediction 1 are relevant as long as it is truly a new discovery of a non-clearly known behavior reported before.

The use of a particular substrate might be related to different other factors rather than a given one, e.g. morphology. Thus, for example, the use of the ground should be tested in relation to predation risk, food availability/location of the food, or male patrolling behavior/intergroup encounters. In section 4.2, it will be relevant to observe these data using a comparative approach with other species of this genus. Likely, this way will serve to explore why the results do not support Prediction 2. In addition, eventually, the position within the canopy structure might influence or serve to explain different uses of the branches by animals of different ages. This can be explored thinking in a reticule that might include different locations within the tree, i.e. bottom to top: upper canopy, middle canopy, and lower canopy as well as center to the exterior: core/near the trunk, middle, periphery. Also, the type, inclination, and diameter of branches should be considered.

In any case, this is a very interesting manuscript and should be published after the issues raised are fully addressed.

Title: It should be more specific. It says: Positional behavior and substrate use in wild macaques

It must say: Positional behavior and substrate use in wild Tibetan macaques

Or

Ontogeny, positional behavior and substrate use in wild Tibetan macaques

Tables: add proper divisions, e.g. highlight the modes in table 3

English language: It should be fully revised. E.g., line 39-40, it says “terrestrial”, it must say “terrestriality”, or line 172, it says “it´s” rather than “its”

Final recommendation: Accept with major revision

Reviewer 2 Report

For the most part this manuscript is well written, and the study design and analysis are appropriate. This is however a very nuanced, topic and the paper fails to argue convincingly that the positional behavior is causal or circumstantial. There is a brief mention of play behavior and anti-predator strategies as an explanation of the difference in substrate use and age, but this is almost a passing reflection as opposed to a major life history situation. This paper could be much more robust, and value would be added by a deeper dive into the socioecology of the species. The discussion could also be improved with clarity and a format that mirrors the intentions set out in the introduction. The authors seem to bounce back and forth between comparison species without a clear description of the niche differences and the relevant socioecology influences for the differences. There are a few specific areas in need of review.

Line 54: This sentence is difficult to read and should be turned into two sentences.

Line 57: “…among the seven new world monkeys…” should be clarified to indicate that this is the seven that the cited author evaluated in a region of Surinam. As it reads now, it appears as though there are only seven New World Monkeys in total.

Line 60: The example given is a good one, but isn’t it fair to consider the diet as a fourth factor? As the citied author does.

Lie 104 seems to end in a typo, as it does not make sense.

Table 1 is incomplete and does not represent the vast amount of diversity within species. Listing the habitat type is confusing, as it is simply the type of the study population. Many of these species live in a mosaic of habitats or exploit different habitat types depending on season. It would be more complete if you listed all of the habitats, they are found rather than the one that fits the hypothesis of the relatively few citations for these species.

Line 127: There appears to be a typo

Line 162 and 163 are repeated.

Line 314 appears to have a typo “… not supported prediction 2.”

Line 328: This sentence does not make sense

Reviewer 3 Report

This is an article on positional behavior in Tibetan macaques (Please use the word Tibetan in the title.)

The language in this manuscript needs an editor who speaks English as their first language.  Because of the grammatical errors, it was challenging to review this document and to understand the significance of the findings.

Introduction:

I started keeping track of the grammatical errors, but given their number, I stopped doing this at the Introduction.  However, here are a couple of examples:

Lines 30-31:  “…significantly more significant…”

Line 32:  “Adult males were more frequent quadrupedal walking…”

If you plan to resubmit, please have someone edit this manuscript.

Methods:

2.3 Behavioral Data collection: 

Data were collected using a scan sample, but it is unclear how the animals were scanned.  Did you have a randomized list and scan each individual in the order on that list?  Did you scan left to right only the animals that were visible?  This needs more explanation.

It is mentioned that a pre-experiment was conducted to ensure accurate recording by field observers.  However, there is no mention of reliability checks being conducted or what the results were.  This is critical to ensure data accuracy.

Table 2.

The ethogram needs to have mutually exclusive categories.  What is the difference between traveling and quadrupedal walk?   What is the difference between rest and lie?  If you are doing a scan sampling, how can you determine whether an animal had not obtained food within 5 seconds?

Results

Table 3- This would be better as a figure

Discussion

In large part due to the grammar and language issues, I have a hard time understanding the significance of the findings and of this manuscript.

For example, in paragraph 1, why does the fact that adult male Japanese macaques spend a similar amount of time terrestrial support the prediction that Tibetan macaques are terrestrial?  What percentage of time on the ground is necessary before an animal is considered to be terrestrial?  More thought needs to go into the conclusions and the support of those conclusions.

Other comments:

Line 297-  “That was the case in two species.”  What two species?

Lines 328-329:  “And the suspension movement, cling movement and dropping in play without analysis.”  What does this mean?

Lines 348-349-  “…the substrate significantly limited the Tibetan macaque’s body weight…”  How does a substrate limit body weight?

Round 2

Reviewer 1 Report

Final recommendation: Accept

Reviewer 2 Report

The corrections and edits made by the authors are sufficient, and I would recommend this manuscript for publication.

Reviewer 3 Report

This manuscript is much improved over the previous version.  I just have a few comments:

page 2, line 112- "limbs are accelerated"  What does this mean?

page 2, line 129- "wood and scrubs"  This would be better as "woodland and scrubland" 

Methods- Were reliability checks conducted?  If so, please provide the outcome.  If not, that makes the data somewhat questionable.

Page 4, line 509- Change "on the move" to "traveling."  The word "traveling" is what is used in the ethogram.

Table 1- Ethogram:

Social grooming:  "two or more" not "two and more"

Social grooming:  Please do not use the same word (grooming) in the definition.

Leap:  What is meant by a "longer stay in the air"?  Longer than what?

Quadrupedal, bipedal stand, lie, suspension:  All definitions say "substrate most of the body" or "substrate the body."  Replace the word "substrate" with "support".

Ground:  "Use the ground as a substrate"  Use a word other than "ground" in the definition.

Page 8, line 686 "...significantly more frequent..." should be "significantly more likely"

Page 9, line 745:  "characterized had classified"  Delete "characterized" or "had classified."

Page 9, line 755-756:  What do you mean by "positional sequence"?

Page 10, line 829:  "...extends the body's extension range..." Delete the word "extension"- "...extends the body's range..."

Page 11, line 897:  "scrub brush" instead of "scrubs"
